# Bile Acids and Biliary Fibrosis

**DOI:** 10.3390/cells12050792

**Published:** 2023-03-02

**Authors:** Sayed Obaidullah Aseem, Phillip B. Hylemon, Huiping Zhou

**Affiliations:** 1Stravitz-Sanyal Institute for Liver Disease & Metabolic Health, School of Medicine, Virginia Commonwealth University, Richmond, VA 23298, USA; 2Division of Gastroenterology, Hepatology and Nutrition, Department of Internal Medicine, Medical College of Virginia, Virginia Commonwealth University, Richmond, VA 23298, USA; 3Department of Microbiology and Immunology, Virginia Commonwealth University, Richmond, VA 23298, USA; 4Central Virginia Veterans Healthcare System, Richmond, VA 23249, USA

**Keywords:** biliary fibrosis, bile acids, bile acid receptors, cholangiocytes, cholangiopathies

## Abstract

Biliary fibrosis is the driving pathological process in cholangiopathies such as primary biliary cholangitis (PBC) and primary sclerosing cholangitis (PSC). Cholangiopathies are also associated with cholestasis, which is the retention of biliary components, including bile acids, in the liver and blood. Cholestasis may worsen with biliary fibrosis. Furthermore, bile acid levels, composition and homeostasis are dysregulated in PBC and PSC. In fact, mounting data from animal models and human cholangiopathies suggest that bile acids play a crucial role in the pathogenesis and progression of biliary fibrosis. The identification of bile acid receptors has advanced our understanding of various signaling pathways involved in regulating cholangiocyte functions and the potential impact on biliary fibrosis. We will also briefly review recent findings linking these receptors with epigenetic regulatory mechanisms. Further detailed understanding of bile acid signaling in the pathogenesis of biliary fibrosis will uncover additional therapeutic avenues for cholangiopathies.

## 1. Introduction

Biliary fibrosis is the defining and predominant pathological feature of many diseases of the biliary system. Diseases affecting the biliary system and cholangiocytes, the epithelial cells that line the biliary tree, are collectively called cholangiopathies [1]. These are a diverse group of diseases that together amount to considerable clinical significance. Fibrogenic cholangiopathies, which are highlighted by biliary fibrosis, include primary biliary cholangitis (PBC) and primary sclerosing cholangitis (PSC). Both PBC and PSC are well-known clinical entities and subjects of intense clinical, translational and basic science research. PSC has a variable but progressive course of biliary fibrosis, eventually culminating in cirrhosis and its associated complications [2]. There are no currently approved pharmaceutical treatments for PSC, which may be ultimately treated with liver transplantation. PBC is treated with the secondary bile acid ursodeoxycholic acid (UDCA), which has been shown to improve the disease’s clinical course, including transplant-free survival [3]. However, in up to 40% of patients, PBC can be progressive despite UDCA treatment. Obeticholic acid is a synthetic bile acid approved for the treatment of PBC patients with inadequate response or intolerant to UDCA.

Fibrogenic cholangiopathies can present clinically with significant cholestasis, particularly with progressive biliary fibrosis [3]. Cholestasis is the retention of bile acids and other bile components within the liver and serum of afflicted patients either due to an inability to secrete or an obstruction [4]. Cholestasis is a significant cause of morbidity in these conditions. Pruritus, one of the chief and debilitating symptoms, is thought to be directly related to the retention of biliary components [5]. However, bile acid retention and dysregulation may not simply be a cause for symptoms. More recent studies suggest a role for bile acids in both the progression of biliary fibrosis and the treatment of these conditions. In this review, we will highlight the role of bile acids in biliary fibrosis with a focus on the effects of bile acid on cholangiocytes.

## 2. Cholangiopathies

Cholangiopathies have diverse etiologies, which include autoimmunity (PBC), genetic (Alagille syndrome), infectious (AIDS cholangiopathy, parasitic infections), malignancy (cholangiocarcinoma), ischemia (ischemic cholangiopathy in post-liver transplant) and idiopathic (PSC, biliary atresia) [1,4]. Regardless of the type of insult to the biliary tree, chronic injury to cholangiocytes and bile ducts can lead to progressive biliary fibrosis (Figure 1). This is initiated with chronic insult causing the cholangiocytes to transform into a proliferative and secretory state. It is accompanied by the recruitment of inflammatory and immune cells [4]. Bile duct proliferation, along with an inflammatory response, is termed ductular reaction (Figure 1). Paracrine and autocrine signals allow interactions between cholangiocytes, immune cells and myofibroblasts. Cholangiocytes release paracrine signals that activate hepatic stellate cells (HSCs) and portal fibroblasts into myofibroblast-type cells [4]. The current paradigm suggests that it is these activated HSCs and portal fibroblasts that lay down the extracellular matrix (ECM) of biliary fibrosis. Probably later in the course of disease, cholangiocytes undergo senescence, a state of cell cycle arrest but highly secretory [6]. The mismatch of bile duct injury and cholangiocyte senescence results in ductopenia, a phenomenon observed later in the disease [4]. With the persistent injury, biliary fibrosis progresses to cirrhosis, ultimately to decompensation with portal hypertension and its associated manifestations, and increased risk of hepatobiliary malignancy.

## 3. Bile Acids and Cholangiocytes

Bile acids play important roles in regulating the lipid, cholesterol, fat, and fat-soluble vitamin trafficking and absorption through bile acid micelle formation [7]. The de novo biosynthesis of bile acids also functions as a cholesterol catabolic pathway [8]. Hepatocytes express all the necessary enzymes for bile acid synthesis, which is carried out by the classical (neutral) and alternative (acidic) pathways. The classical pathway, which is responsible for the majority of bile acid synthesis, converts cholesterol into cholic acid (CA) and chenodeoxycholic acid (CDCA). Cholesterol-7 alpha-hydroxylase (CYP7A1) is the rate-limiting enzyme for this biochemical pathway. The alternative pathway produces mostly CDCA. These primary bile acids are then conjugated mostly with glycine in humans, but also with taurine. Conjugated bile acids are secreted into bile ducts where they can be picked up by cholangiocytes through the action of apical sodium-dependent bile acid transporter (ASBT), also known as ileal bile acid transporter (iBAT), and released into the peribiliary plexus for delivery back to the hepatocytes, creating the cholehepatic shunt [8,9,10]. Bile acids not taken up by cholangiocytes are stored in the gallbladder and then released into the intestine after food intake, where they have essential functions in nutrient absorption. Bile acids are efficiently reabsorbed in the small intestine by passive mechanisms and active mechanisms in the terminal ileum by ASBT, such that up to 95% of secreted bile acids are reabsorbed [11]. A small fraction of bile acids undergo further modification by the gut microbiome: first deconjugation, followed by various other transformation reactions, including the oxidation of α- hydroxyl groups at C-3, C-7 and C12 to form oxo groups and the reduction of these to form β-hydroxyl groups, yielding 3-iso, 7-epi and 12-epi bile acids. In addition, selective species of the genus *Clostridium* carry out 7α-dehydroxylation of the primary bile acids, CA and CDCA, forming the secondary bile acids, deoxycholic acid (DCA) and lithocholic acid (LCA), respectively [11]. There are over 50 bile acid metabolites identified in humans [12]. Secondary bile acids can also be reabsorbed by the gut and transferred to the serum, followed by reuptake in the liver. Bile acid secretion into the intestine and reabsorption back to the liver via the portal vein is known as enterohepatic circulation [11].

Cholangiocytes are polarized epithelial cells that line the biliary tree. The apical membrane has microvilli and a primary cilium that responds to mechanical, chemical and osmolar stimuli to maintain cholangiocyte homeostasis [13]. Smaller intrahepatic bile ducts are lined by small, cuboidal cholangiocytes, whereas larger ducts are lined by large, columnar cholangiocytes. There is likely a significant variety and a spectrum in the functions, gene expression, and role in the pathology of cholangiocytes rather than the simple categorization of small versus large [14]. Regardless, both small and large cholangiocytes express various bile acid transporters. ASBT can facilitate the transport of bile acids from the bile into cholangiocytes. At the basolateral membrane, organic solute transporter (OST) alpha and beta and multidrug resistance-associated protein 3 (MRP3) can transport bile acids into the systemic circulation [15]. These, along with several other electrolyte transporters with both absorptive and secretory functions and aquaporins, modulate the biliary composition of bile acids, electrolytes and water, thereby carrying out an important function in the ultimate content of the bile secreted into the duodenum [15]. Importantly, cholangiocytes secrete bicarbonate at the apical membrane through the anion exchanger 2 (AE2) to form the so-called “bicarbonate umbrella” [16]. This is thought to protect cholangiocytes from toxic substances in the bile, including certain bile acids that are prevented from diffusing into cholangiocytes by maintaining them in their anion form by the alkaline lumen. Disruption of the bicarbonate umbrella is associated with certain pathological conditions.

### 3.1. Bile Acids in Fibrogenic Cholangiopathies

Recent studies have examined the bile acid composition in cholangiopathies, mostly PBC and PSC. Most studies have focused on the serum bile acid composition as a surrogate of the bile acid pool in the enterohepatic circulation. There is also a more limited number analyzing the bile acid composition in bile obtained by endoscopic means.

PBC patients were shown to have increased total serum bile acid levels with predominantly increased primary bile acids in a study by Chen et al. Serum secondary bile acids were reduced and there were no identifiable differences in the fecal bile acid levels [17]. Mousa et al. showed that a cohort of PSC patients similarly had increased total and conjugated bile acids. An increased ratio of primary to secondary bile acids was also noted in this PSC cohort. Hepatic decompensation was associated with the increased concentration and conjugated fraction of many bile acids, but the glycine to taurine conjugation ratio appeared protective [18]. It is worth noting that most of the PSC patients in this study had decompensated disease (80–100%). Another study of urinary bile acid levels, revealed elevated levels in most liver disease patients, with PBC and PSC showing the highest total levels and PSC showing the highest CA and CDCA levels [19]. In contrast, the total fecal bile acids were shown to be reduced in PSC with IBD compared to those without PSC [20]. An earlier study of both PBC and PSC cohorts analyzing 17 bile acids in the serum showed that the concentrations of total bile acids and taurine and glycine conjugates of primary bile acids were increased in both cohorts [21]. Secondary bile acids were reduced in the PSC cohort. The PBC cohort, however, did not show a reduction in secondary bile acids, contradicting the study by Chen et al. [17,21]. These inconsistencies are possibly due to cohort differences, particularly the stage of disease, such as non-cirrhotic, cirrhotic and decompensated cirrhotic. A more recent study of largely non-cirrhotic patients found that the bile fluid concentrations of most bile acids were reduced in PSC except for taurolithocholic acid, a noxious secondary bile acid [22], consistent with a previous study [23]. One obvious justification for the lower bile acids in the bile fluid in contrast to the elevated bile acids in the serum shown by other studies may be that, due to the PSC-related dysfunction, the liver has a reduced capacity for secreting bile acids into the bile fluid, resulting in cholestasis. However, this assumption cannot be maintained without a concomitant analysis of serum bile acids and can be confirmed in future studies simultaneously showing lower bile acids in bile fluid and higher serum bile acids. Other possibilities include leakage from the biliary tree or dilution, but the biliary bile acid concentration appears to be intact [23,24]. In cirrhosis, regardless of etiology, the bile acid pool may be depleted due to decreased synthesis and disproportionate partitioning into the liver and serum due to poor hepatic secretion [25,26]. Therefore, further analyses are required to determine how much of the serum and bile fluid bile acid levels are related to advanced liver disease versus etiologically to specific cholangiopathies.

### 3.2. Bile Acids and Biliary Fibrosis

Bile acids have been directly and indirectly implicated in biliary fibrosis and various animal models of biliary fibrosis have been used to identify potential mechanisms. Fibrosis is a response to injury and inflammation. Bile acids, owing to their detergent properties, were previously considered cell-injurious by solubilizing the plasma membrane. However, the effects of bile acids on cholangiocytes and other hepatic cells may be much more complex [15]. Certain bile acids, such as glycochenodeoxycholic acid (GCDCA) and lithocholic acid (LCA), may cause necrotic cell death and lead to an inflammatory response [27,28]. However, the pathophysiological relevance of these observations is unclear since most LCA is conjugated or bound to serum albumin/lipoproteins [29,30]. Generally, hydrophobic bile acids are more likely to be harmful (such as with the LCA feeding model) and hydrophilic ones (UDCA/norUDCA) are more likely to be protective [28]. Administering hydrophilic bile acids to mouse models of biliary injury and fibrosis was protective [31]. However, bile duct injury, inflammation and associated fibrosis may not be solely attributable to hydrophobic bile acids, as taurocholic acid (TCA) has also been implicated in cholestatic injury and fibrosis [32]. Despite these lines of evidence, controversy remains as to whether bile acids have direct injurious effects on cholangiocytes. At pathophysiological concentrations, bile acids resulted in a cytokine-induced inflammatory response in hepatocytes only [33]. Furthermore, limited studies in human cholangiopathies thus far have not revealed significant changes in the composition of biliary bile acids to suggest toxicity to cholangiocytes [22,34]. Larger, high-quality studies of biliary or duodenal bile acid composition in cholangiopathies are needed to provide further insight.

Bile acids may become noxious in the absence of protective mechanisms such as impaired phospholipid secretion. This is supported by the seminal study of multidrug resistance protein 2 (Mdr2) gene knockout in mice resulting in impaired biliary phospholipid secretion with consequent bile-acid-induced biliary duct injury [35]. The biliary injury leads to an inflammatory response and the activation of hepatic myofibroblasts, which subsequently mediate biliary fibrosis [36]. Further support comes from the study of a genetic disorder, progressive familial intrahepatic cholestasis type 3 (PFIC3), where mutations in MDR3 markedly reduce biliary phospholipid levels. This condition is associated with cholangiocyte injury, ductular reaction, inflammation and fibrosis [37]. Analyses of biliary phospholipids in PSC thus far have not revealed any changes compared to controls to suggest such a mechanism at play [23,24]. Another potential path to the noxious effects of bile acids may be through the disruption of the protective bicarbonate umbrella and transporters such as AE2 involved in its maintenance. Indeed, bile acid toxicity is increased in a pH-dependent manner, especially with the disruption of a cholangiocyte apical glycocalyx [38]. Congruently, reduced AE2 expression has been noted in PBC patients [39]. In contrast, canalicular and basolateral efflux pumps may be upregulated, while basolateral uptake systems are downregulated in adaptive changes to curb bile acid accumulation within cholangiocytes [40].

Biliary fibrosis may involve mechanisms without direct injury to cholangiocytes. Increased serum bile acids may directly activate hepatic myofibroblasts to lay down the ECM of biliary fibrosis. For example, CDCA treatment of a hepatic stellate cell (HSC) cell line LX2 resulted in proliferation and collagen deposition, suggesting activation [41]. Conjugated secondary 12a-hydroxylated bile acids, taurodeoxycholate (TDCA) and glycodeoxycholate (GDCA), found to be significantly increased in non-alcoholic steatohepatitis (NASH) patients and mouse models, activated LX2 cells much more than other bile acids tested [42]. However, elevated serum bile acids of normal composition are unlikely to directly activate HSCs to produce fibrosis. This notion is supported by the fact that pediatric patients with mutations and deficiency in the Na+–taurocholate cotransporting polypeptide (NTCP, *SLC10A1*), the major transporter of conjugated bile salts from the plasma compartment into the hepatocyte, have markedly elevated conjugated serum bile acids but with normal liver function and without signs of injury or fibrosis [43]. Most other inborn bile acid transporter defects (such as PFIC2/3, Alagille syndrome and cystic fibrosis) present with injury to hepatocytes, cholangiocytes or both and associated inflammation that leads to hepatobiliary fibrosis [15]. Furthermore, if serum bile acids directly activated HSCs, a perisinusoidal fibrosis pattern would be the more typical pattern expected. In contrast, in fibrogenic cholangiopathies, such as PSC and PBC, fibrosis starts in a periductal manner that eventually propagates to cirrhosis. Therefore, while the direct activation of HSCs by serum bile acids is plausible, it is unlikely to be the primary mechanism of biliary fibrosis in fibrogenic cholangiopathies, especially in the earlier stages.

Bile acids may play a role in other aspects of biliary fibrosis in cholangiopathies, namely cholangiocyte proliferation, ductular reaction and cholangiocyte senescence. These processes will be reviewed along with the function of bile acid receptors in the following sections.

### 3.3. Bile Acid Receptors and Biliary Fibrosis

#### 3.3.1. G-Protein-Coupled Bile Acid Receptor 1 (GPBAR1) (Also Known as Transmembrane G-Protein-Coupled Receptor 5 (TGR5))

GPBAR1 (TGR5) is expressed in all hepatic cells except hepatocytes [44]. It is mainly activated by secondary and unconjugated bile acids (LCA>DCA>CDCA>CA) (Table 1). GPBAR1 has been shown to localize in multiple cholangiocyte compartments, but ciliary-membrane-bound GPBAR1 appears to have pronounced effects [45]. In vitro experiments to determine GPBAR1 signaling from different compartments utilized the phenomenon of early confluent cholangiocytes that do not form a cilium until an additional 7 days of confluence [45]. Ciliated cholangiocytes had reduced cAMP but elevated extracellular regulated protein kinase (ERK) activation and suppressed proliferation (Figure 2A). Non-ciliated cholangiocytes had the opposite cAMP and ERK phenotype and activated proliferation with GPBAR1 activation (Figure 2A) [45]. Structurally or functionally defective cilia result in several cholangiopathies (e.g., polycystic liver disease and biliary atresia) characterized by cholangiocyte hyperproliferation [46,47], possibly through apical GPBAR1 activation with increased intracellular cAMP [45,48]. Mouse models of biliary fibrosis and cholestasis showed an increased cholangiocyte proliferation via the Gpbar1-mediated activation of the epidermal growth factor receptor and ERK pathway [49]. Gpbar1-knockout mice had reduced cholangiocyte proliferation with bile duct ligation (BDL), which causes intrahepatic biliary fibrosis presumably through effects on apical Gpbar1 activation (Figure 2A). These findings point to a deleterious role for GPBAR1 in cholestasis, where bile acid activation of GPBAR1 may stimulate small-cholangiocyte proliferation, a necessary component of the ductular reaction, which ultimately progresses to biliary fibrosis. This notion is supported by other lines of evidence in different mouse models. Gpbar1 deletion in NASH- and CCl4-induced injury mouse models led to reduced hepatic fibrosis. The fibrogenic role of Gpbar1 was proposed to be through bile acid-induced HSC activation via the ERK/p38 MAPK signaling pathways [42]. Currently, there are no GPBAR1 antagonists being studied in biliary fibrosis. However, a first-in-class GPBAR1 antagonist, designated SBI-319, was recently demonstrated to inhibit cholangiocyte proliferation and cystogenesis in models of polycystic liver disease (T. Masyuk et al. unpublished data presented at the Liver meeting parallel 36 section). 

In contrast, other lines of evidence do not support such a deleterious role for GPBAR1. Firstly, GPBAR1 levels are reduced in PBC and PSC patients [50]. GPBAR1 agonists promote bile flow [51] and may have antiapoptotic effects in cholangiocytes [52,53], which would be protective in cholangiopathies. In a mouse model of cholangiopathy, dual FXR and GPBAR1 agonists improved biliary fibrosis, but not selective GPBAR1 or FXR agonists [54]. Furthermore, GPBAR1 appears to attenuate liver fibrosis in a diabetic mouse model [55]. Some of these contradictory observations may be explained by cell- and injury-specific effects of GPBAR1. Consistently, bile-acid-mediated GPBAR1 activation in Kupffer cells induced pro-inflammatory cytokines via the c-Jun N-terminal kinase (JNK) pathway (Figure 2A) [56]. The bile acid induction of cytokine production in Kupffer cells may suppress hepatic Cyp7a1 in a negative feedback loop [57]. Conversely, GPBAR1 activation in monocytes and macrophages inhibited nuclear factor kappa B (NF-κB) and subsequently pro-inflammatory cytokine production in a model of atherosclerosis [58]. Therefore, GPBAR1 appears to have opposing roles in different cell populations. The pro- and anti-inflammatory cytokine production through different immune cell populations may have a role in propagating or ameliorating biliary fibrosis [59,60]. The effects of these cell-specific observations have not yet been studied in detail in biliary fibrosis models.

#### 3.3.2. Farnesoid X Receptor (FXR)

The FXR nuclear receptor is widely expressed, including in the liver, intestine and immune cells. The activation of FXR by bile acids varies according to species (CDCA>>>DCA>CA>LCA) (Table 1) [61]. FXR, through its tissue-specific effects, is the master regulator of bile acid homeostasis. FXR activation in the liver and intestine leads to gene expression changes to downregulate bile acid de novo synthesis. This is accomplished by the FXR-dependent transactivation of small heterodimer partner (SHP) by phosphorylation and inhibiting proteasomal degradation in hepatocytes [62]. SHP subsequently represses CYP7A1, the rate-limiting enzyme in bile acid synthesis. Intestinal FXR signaling leads to the release of fibroblast growth factor (FGF)15/19 secreted from the ileum, which activates fibroblast growth factor receptor 4 (FGFR4) and ERK signaling to also suppress CYP7A1. The FGFR4 and ERK signaling additionally inhibit SHP proteasomal degradation, which may also suppress CYP7A1 [62]. FXR has immune-modulating effects through interactions with NF-κB in hepatocytes and various immune cells, leading to the suppression of NF-κB-induced inflammatory cytokines [7,63,64]. Conversely, pro-inflammatory cytokines may repress FXR activation by various mechanisms, including NF-κB activation [65].

Under cholestatic conditions, FXR orchestrates the above and other adaptive transporter changes to counteract the damaging effects of cholestasis. These adaptive mechanisms are not restricted to the liver hepatocytes but also occur in the kidney, intestine and cholangiocytes [66]. They include the increased expression of ASBT and OSTα/β in cholangiocytes due to ductular proliferation, which allows bile acid reabsorption from obstructed bile ducts (Figure 2B) [66]. The overall effect is expected to promote bile flow in cholestasis, thereby preventing the hepatic accumulation of toxic bile acids. Animal models, however, have produced conflicting data regarding the role of FXR in cholestasis. Whereas FXR deletion had protective effects, FXR agonists reduced hepatic injury and inflammation in rodent models of cholestasis [67,68]. In humans, FXR expression is increased in both PBC and PSC [69,70]. Yet, the expected FXR-mediated downregulation of CYP7A1 was only observed in PBC patients, while PSC patients had unaltered CYP7A1 expression.

Given the role of FXR in regulating cholestasis and the associated inflammatory response, its activation is expected to reduce biliary fibrosis. Indeed, activation of the FXR-SHP axis in rodent models of cholestasis prevented fibrosis [71]. A direct effect on the FXR expressed in HSCs has been proposed as a potential mechanism for this anti-fibrotic effect. However, HSCs express low levels of FXR [72]. Therefore, cholangiocyte signaling and crosstalk with HSCs may be another target of the effects of FXR signaling [73]. In a rat liver transplant model, biliary transit time and bile duct injury were prolonged with pre-transplant ischemia of the donor liver, which correlated with reduced Fxr expression [74]. Cholangiocytes under hypoxic conditions had reduced expression of Fxr and increased expression of the fibrogenic factor Tgfβ [74]. Loss-of-function mutations in the FXR gene result in PFIC5 consisting of severe cholestasis and rapid progression of fibrosis to cirrhosis [75]. These studies point to the important role of FXR in regulating biliary fibrosis. Further support comes from clinical trials that have provided evidence for the use of FXR agonists in cholestatic liver diseases. Prominently, obeticholic acid is an approved FXR agonist for the treatment of PBC. Long-term treatment has shown stabilization of biliary fibrosis [76]. There are multiple other FXR agonists in the clinical trial stages to treat cholestatic liver diseases [77]. Interested readers are referred to recent review articles on the subject [78,79].

#### 3.3.3. Sphingosine-1 Phosphate Receptor (S1PR) 2

S1PR2 is a member of five S1PRs, originally discovered as endothelial differentiation gene 5 [80]. S1PR2 is expressed in most tissues and associated with different G-proteins specific to cells and stimuli [81]. It is highly expressed in the liver and is the predominant S1PR in hepatocytes and cholangiocytes [32,82]. Conjugated primary bile acids activate S1PR2, with TCA being the most potent activator (Table 1) [82,83]. In hepatocytes, conjugated bile acids activate the extracellular signal-regulated kinase 1 and 2 (ERK1/2) and protein kinase B (AKT) signaling pathways through S1PR2, which upregulates sphingosine kinase 2 (SphK2). S1PR2 and SphK2 appear to have important roles in hepatic lipid and glucose metabolism. Mice deficient in either S1pr2 or Sphk2 develop rapid steatosis on a high-fat diet [83]. In cholangiocarcinoma cell lines, conjugated bile acids activated S1PR2 and upregulated cyclooxygenase 2, which led to invasive growth and bile duct proliferation [84,85]. The TCA-mediated activation of ERK1/2 and AKT in cholangiocytes was inhibited by S1PR2 antagonists or siRNA knockdown, which also inhibited cholangiocyte proliferation and migration (Figure 2C) [32]. In cholestatic mice, S1PR2 expression was increased, and S1PR2 deficiency attenuated cholestasis-mediated cholangiocyte proliferation, cholestatic injury, inflammation and fibrosis (Figure 2C) [32]. Similarly, macrophage-specific knockdown of S1PR2 reduced cholestasis-associated inflammation and fibrosis by attenuating the chronic-liver-injury-associated NOD-like receptor family pyrin domain containing 3 inflammasome [86]. Similar results were observed in the liver fluke (*Clonorchis sinensis*)-infected mouse model of hepatobiliary injury and fibrosis where inflammation, bile duct hyperplasia and periductal fibrosis could be attenuated by an S1PR2 inhibitor [87]. The significance of these findings in human cholangiopathies is yet to be clarified.

#### 3.3.4. Subtype 3 Muscarinic Acetylcholine Receptor (M3R)

Cholangiocytes predominantly express M3R, which is responsible for mediating the effects of parasympathetic innervation but is also activated by TLCA [88]. Parasympathetic effects through the vagus nerve increase bile flow and HCO_3_^−^ secretion, which are inhibited with vagotomy [89]. Similarly, M3R-deficient mice showed reduced bile flow but did not spontaneously develop cholestasis, injury or fibrosis [90]. They were, however, more susceptible to DDC-induced liver injury. Likewise, M3R agonist treatment of Mdr2^−/−^ mice also reduced liver injury, but in neither model was there a substantial effect on biliary fibrosis [90]. These findings suggest that M3R may have a role under cholestatic conditions when the increased bile acid exposure would further activate M3R to increase bile flow and HCO_3_^−^ secretion in order to limit the cholestatic damage. Consistently, denervated transplanted livers have an increased risk of ischemic cholangiopathy, especially with prolonged ischemic time [91]. However, further supportive data from other cholangiopathies are lacking.

#### 3.3.5. Vitamin D Receptor (VDR)

While hepatocytes may express low levels of VDR, cholangiocytes and non-parenchymal hepatic cells express VDR abundantly [92]. Vdr-deficient mice develop spontaneous liver fibrosis, which is proposed to be through ungated effects of the Tgfβ1/Smad activation in HSCs [93]. Consistently, VDR agonists protected against CCL4-induced liver fibrosis [93]. VDR also functions as a receptor for LCA [94]. Vdr deficiency in the BDL cholestasis model resulted in increased liver damage [95]. The damage was partly due to limited adaptive changes in bile acid transporters and partly due to increased bile duct rupture due to EGFR-mediated altered E-cadherin expression [95]. These findings suggest that cholestasis may activate VDR to produce adaptive changes in bile acid transporters as a protective mechanism against the toxic effects of cholestasis. Similar findings were reported in a Vdr/Mdr2^−/−^ double knockout model that had worsened inflammation and fibrosis [96]. Taken together, these lines of observations support bile-acid-related and -unrelated protective roles of VDR in cholangiopathies. PSC and PBC patients frequently have vitamin D deficiency which appears to correlate with disease severity [97,98,99]. VDR polymorphisms and lower expression have been reported in PBC, which may have a role in the progression of the disease [100,101]. Further detailed insight into the expression and phenotype of VDR in PSC and other cholangiopathies is needed for a better understanding of the role of this receptor in fibrosing cholangiopathies [102].

#### 3.3.6. Pregnane X Receptor (PXR) and the Constitutive Androstane Receptor (CAR)

PXR and CAR are highly expressed in the liver, followed by the intestine [103]. They act as sensors for both exogenous and endogenous toxic products to signal detoxification and metabolism [104]. These receptors may also have a role in liver disease. Both receptors form heterodimers with retinoid X receptor for downstream signaling [103]. PXR is activated by LCA and its derivatives [105]. Its activation appears to protect against LCA- or BDL-induced liver toxicity by regulating bile acid synthesis and transporters [105,106,107]. It may also have a role in regulating inflammation arising from hepatocytes via NF-κB signaling [107]. CAR may have complementary roles to PXR and FXR. In a PXR and FXR double knockout model, CAR expression was increased, and its activation was protective by the modulation of genes involved in bile acid and bilirubin metabolism [108]. Yet, genetic deletion of PXR, CAR or both demonstrated that CAR deficiency led to more severe liver toxicity and that the protective mechanisms of PXR and CAR were through the modulation of different enzymes and transporters [109]. PXR may be increased in PSC, but its protective effects may be tampered by the attenuation of its target genes through other mechanisms [110]. Despite these lines of evidence, the role of PXR or CAR in biliary fibrosis remains to be fully evaluated. Furthermore, it is not clear if diseased cholangiocytes express PXR or CAR in cholangiopathies.

#### 3.3.7. Retinoic-Acid-Related Orphan Receptor γt (RORγt)

RORγt is a nuclear receptor expressed in immune cells where it has an important role in cell maturation and differentiation, particularly in regulating the pro- and anti-inflammatory homeostasis of T helper (Th) 17 and other subsets of CD4^+^ Th cells [111]. RORγt has been implicated in inflammatory and autoimmune diseases, including inflammatory bowel disease (IBD) [112]. RORγt is directly bound by the secondary bile acid 3-oxo-LCA, which can modulate RORγt transcriptional activity [113]. Several studies have implicated microbiome-generated secondary bile acids in RORγt-regulation of Th17 and other CD4+ cells in IBD, which may be either through directly regulating RORγt transcriptional activity or indirectly via other bile acid receptors [111,113,114]. RORγt inverse agonists potentially have anti-inflammatory properties by modulating IL-17a/f levels and may be an attractive target for autoimmune disease [111]. IL-17a may activate HSCs [115]. BDL mouse models have demonstrated an increased expression of IL-17, TGF-β and RORγt [116]. Consistently, RORγt knockdown in a hepatocyte-injury mouse model reduced the hepatocyte epithelial–mesenchymal transition and ameliorated liver fibrosis [117]. These studies suggest that RORγt may have a fibrogenic role in the liver with its effects on hepatocytes. Whether RORγt is also expressed in cholangiocytes or has a role in biliary fibrosis has not been studied in detail.

## 4. Bile Acids, Microbiome and Biliary Fibrosis

Gut microbiota deconjugate a small portion of bile acids through their bile salt hydrolases (BSH). Further transformation reactions include hydroxyl group oxidation, epimerization and 7α/7β-dehydroxylation forming secondary bile acids. Secondary bile acids can be reabsorbed by the gut by passive transport and transferred to the serum followed by reuptake in the liver. The gut microbiota is known to be altered in cholangiopathies [118]. For example, in PSC, microbial diversity is reduced while certain groups such as Enterobacteriaceae, *Enterococcus* and *Veillonella* are over-represented [118]. Members of the family Enterobacteriaceae contain very potent lipopolysaccharides (LPS) that may contribute to the enhanced inflammatory response in the liver. Gut dysbiosis affects not only bile acid metabolism but also intestinal permeability, short-chain fatty acid availability and the metabolism of macromolecules. These processes can combine to affect dietary energy utilization, inflammation, and liver injury [118].

Germ-free mice demonstrate an increased bile acid uptake from the gut [119]. Germ-free conditions were also shown to result in worsened biliary injury and fibrosis in the Mdr2^−/−^ model of biliary fibrosis [120,121]. There are several possible explanations for this observation. Germ-free conditions were shown to increase the expression of the rate-limiting enzyme in the synthesis of bile acids, Cyp7a1, while also increasing and expanding the small intestinal expression of Asbt [119]. These effects combined to increase the serum and hepatic bile acid content [119], which is one explanation for the toxicity of germ-free conditions. Similarly, antibiotic treatment to obtain germ-free Mdr2^−/−^ mice demonstrated an increased hepatic bile acid concentration and disruption of the bile duct barrier function with resultant bile duct injury [122]. Both processes were shown to be dependent on the reduced bile acid activation of FXR under germ-free conditions [122]. In contrast, the opposite effects were observed in NOD.c3c4 mice, which are a model of immune-mediated biliary injury. Germ-free or antibiotic-treated NOD.c3c4 showed reduced biliary injury, although fibrosis was not affected [123]. These seemingly conflicting observations are likely due to the mouse models used, with NOD.c3c4 being susceptible to microbiota, likely because of its autoimmune predilection. In contrast, the Mdr2^−/−^ studies point to the important role of gut microbiota in bile acid homeostasis. However, neither model addresses the dysbiosis observed in cholangiopathies.

A more elegantly designed study evaluated the effect of inoculating gnotobiotic mice with PSC-derived microbiota. These mice showed a Th17-cell response in the liver and an increased susceptibility to DDC-induced hepatobiliary injury and fibrosis [124]. The role of bile acids in this model was not studied in detail. Other studies have pointed out the deleterious effects of bacterial strains in these models. *E. faecalis* and *E. coli* strains accelerated inflammation and mortality, while Lachnospiraceae colonization of Mdr2^−/−^ mice reduced fibrosis, inflammation and translocation of pathobionts by producing short-chain fatty acids [121]. *K. pneumoniae* enrichment in PSC microbiota may disrupt the epithelial barrier to initiate bacterial translocation and liver inflammatory responses [124]. Similarly, bile duct colonization with Enterococci conferred a risk of disease progression in PSC [125]. It is important to point out that bile acids and the microbiome have an inter-dependent relationship with both exerting effects on the other. Changes in the composition of bile acids due to cholestatic disease may also modulate the microbiome into a deleterious phenotype [126,127]. Consistently, fecal transfer from cholestatic Mdr2^−/−^ mice into wild-type mice induced significant liver injury along with NLRP3 inflammasome activation and cholestasis [126]. Further careful studies in these models are required to determine the effects of dysbiosis on bile acid species, homeostasis and bile acid effects on biliary fibrosis.

## 5. Bile Acids and Epigenetics of Biliary Fibrosis

Epigenetics are reversible, heritable processes that regulate gene expression and determine the cell phenotype without altering the genomic DNA sequence. These processes may be classified into DNA modifications such as methylation, non-coding RNA-mediated gene modulation, histone modifications, and chromatin organization/remodeling [128,129]. These mechanisms have overlapping responses to diverse stimuli and combine to affect gene expression. Various histone-modifying enzymes with their respective modifications have been reported to regulate bile acid metabolism [130,131,132]. Of note, FXR-mediated regulation of bile acid transporters, BSEP, multidrug resistance-associated protein 2 (MRP2) and NTCP recruits the coactivators mixed lineage leukemia 3 (MLL3) or MLL4 to the promoters for histone H3 lysine 4 methylation, which allows for gene expression of these transporters [133,134]. More recently, microRNA-210 was reported to downregulate MLL4 and consequently the FXR target genes, BSEP and SHP. Silencing of microRNA-210 in mice attenuated bile-acid-induced liver injury through MLL4. Cholestatic mice and PBC subjects were shown to have increased microRNA-210 and reduced MLL4 expression [135]. Multiple other non-coding RNAs and epigenetic modifications have been implicated in the pathogenesis of PBC [136,137,138]. Similarly, epigenetic and epigenomic changes have been implicated in the pathogenesis of PSC [128,139,140] including those mediating the interactions between cholangiocytes and hepatic myofibroblasts [141,142,143,144]. 

The role of bile acids in the epigenetic regulation of biliary fibrosis remains an understudied area. Long noncoding RNA H19 (lnRNA-H19) has been reported to be markedly elevated in an Mdr2^−/−^ mouse model and PSC [145]. LnRNA-H19 may be released from cholangiocytes in exosomes. An lnRNA-H19 deficiency significantly protected mice from liver fibrosis in BDL and Mdr2^−/−^ mice along with reduced HSC activation by cholangiocyte-derived, lnRNA-H19-deficient exosomes [146]. It was further demonstrated that lnRNA-H19 deficiency protects mice from BDL-induced cholangiocyte proliferation and biliary fibrosis [147]. The lnRNA-H19-mediated cholangiocyte proliferation was shown to be through the bile-acid-induced expression and activation of S1PR2 and SphK2 [147]. An additional mechanism includes lncRNA-H19-mediated macrophage activation and associated inflammation under cholestatic conditions [148]. These observations indicate the direct and indirect roles of bile-acid-induced lnRNA-H19 in regulating biliary fibrosis. Further studies are required to examine the direct role of bile acid species in regulating other epigenetic mechanisms of biliary fibrosis in detail.

## 6. Conclusions and Future Directions

Biliary fibrosis is the predominant pathological process in cholangiopathies, responsible for the progression to advanced liver disease and associated complications, including the risk of hepatobiliary malignancy. Cholangiopathies are also defined by cholestasis, in which bile acids accumulate in the liver and serum compartments. Cholestasis may worsen with progressive biliary fibrosis. Animal models and human genetic diseases have demonstrated a clear role for bile acids in the pathogenesis and progression of biliary fibrosis. While previously possible direct cytotoxic properties of some bile acids were implicated in cholestatic disease, it is now evident that bile acids primarily acting through receptors and signaling pathways impose their damaging and protective effects. Cholangiocytes, the targets of disease in cholangiopathies, have a role in normal bile acid homeostasis but are also affected by cholestasis. Bile acids, through the activation of GPBAR1 (TGR5) or S1PR2, result in cholangiocyte proliferation, driving in part the ductular reaction, and may be associated with biliary fibrosis. This process presumably amplifies the cholangiocyte production and secretion of fibrogenic signals to activate hepatic myofibroblasts, which produce the ECM of biliary fibrosis. The bile-acid-induced activation of FXR, VDR, M3R, PXR or CAR may lead to protective effects. FXR activation in cholangiocytes leads to modifications in bile acid transporters as adaptive changes to protect against cholestasis. FXR may also attenuate inflammation, indirectly protecting cholangiocytes from inflammatory damage. Combined, these effects of FXR activation may attenuate biliary fibrosis. Similarly, VDR activation under cholestatic conditions may also be protective against cholangiocyte injury and fibrosis. M3R, PXR and CAR are protective against cholangiocyte injury, but either do not attenuate fibrosis or have not been evaluated in detail for biliary fibrosis. RORγt, another bile acid receptor, may have anti-fibrotic effects through its regulation of immune cells and transition to a mesenchymal phenotype, but this receptor has not been studied in cholangiocytes in detail. Similarly, further details of the signaling pathways of bile acid receptors, including epigenetic activators and repressors, have not been explored in detail.

Despite the advancements in understanding bile acid signaling through several receptors and potential roles in cholestasis, controversies and conflicting observations remain in understanding the full picture of bile acid signaling in cholestatic disease models. GPBAR1 (TGR5) appears to have cell-specific effects that can either promote or antagonize components of cholestatic liver injury. Similarly, while there is mounting clinical evidence supporting the beneficial role of FXR agonism in cholestatic liver diseases, surprisingly, FXR deletion had protective effects in rodent models of cholestasis. Further light can be shed on these controversies by lineage- and cell-specific deletion of these receptors in cholestatic animal models. Cholangiocyte-specific deletion of bile acid receptors may reveal their roles in cholestasis and biliary fibrosis in greater detail. Similarly, single-cell and spatial transcriptomic technologies may reveal greater detail in the bile-acid- and bile-acid receptor-specific signaling under cholestatic conditions.

In clinical research, preclinical observations are increasingly applied to further therapeutic options for the two most common cholangiopathies, PSC and PBC. USDA, its derivatives and FXR agonists are studied in completed or ongoing clinical trials. Bile acid levels are altered in these diseases in the serum and stool. Yet, bile acid exposure to cholangiocytes, i.e., bile acid in the bile ducts, has not been examined in great detail. Further studies in larger cohorts would identify changes in bile acid levels as well as the changes in specific bile acid species in these conditions that would be stimulating cholangiocytes. Similarly, the interactions of the microbiome and bile acids and vice versa have to be studied in greater detail. Given the differences between murine and human bile acids, murine models with a humanized bile acid pool may serve an important role in addressing these questions.

## Figures and Tables

**Figure 1 cells-12-00792-f001:**
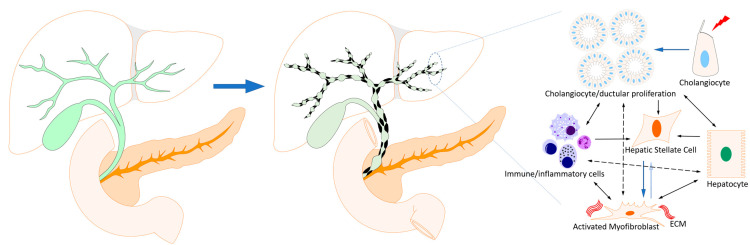
Cell-to-cell interactions in and mechanism of biliary fibrosis. Biliary fibrosis in PSC results in the scarring of normally smooth intra- and extrahepatic bile ducts that transform into a “beaded” appearance. Mechanistically, injury to cholangiocytes transforms them into a proliferative and highly secretory phenotype. Bile ducts proliferate. Immune/inflammatory cells interact with injured cholangiocytes, which together perpetuate a cycle of inflammation. Signals emanating from cholangiocytes and immune cells transform hepatic stellate cells and portal fibroblasts into activated myofibroblasts that secrete the extracellular matrix (ECM) of biliary fibrosis.

**Figure 2 cells-12-00792-f002:**
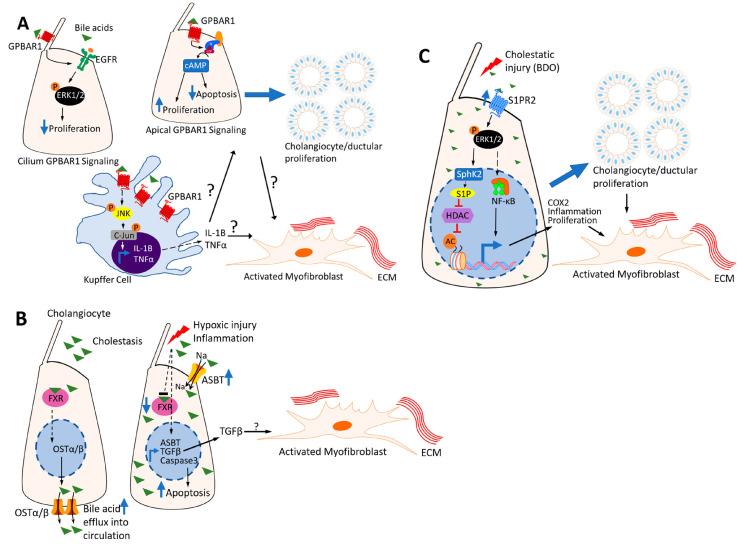
The role of bile acid receptors in biliary fibrosis. (**A**) Bile acid activation of GPBAR1 (TGR5) in the cilium compartment suppresses proliferation but activation of apical GPBAR1 promotes proliferation in cholangiocytes. In Kuppfer cells, GPBAR1 (TGR5) activation stimulates expression of inflammatory cytokines, which may activate myofibroblasts. (**B**) FXR increases the expression of OSTα/β to facilitate the efflux of bile acids under cholestatic conditions. With hypoxic injury or overwhelming inflammation, FXR expression/activation is decreased with increased expression of ASBT, apoptotic machinery and TGFβ, which may activate myofibroblasts. (**C**) S1PR2 expression is increased with cholestatic injury. Bile acid activation of S1PR2 activates ERK1/2 signaling, which promotes sphingosine kinase 2 (SphK2)-mediated sphingosine 1-phosphate levels (S1P). S1P promotes gene expression by inhibiting histone deacetylases (HDACs). ERK1/2 also promotes inflammation and proliferation in part through NF-κB signaling.

**Table 1 cells-12-00792-t001:** Selected bile acid receptor expression in the liver and activation by bile acids.

	GPBAR1 (TGR5)	FXR	S1PR2
Hepatocytes	NO	Yes	Yes
Kupffer cells	Yes	Yes	Yes
Cholangiocytes	Yes	Yes	Yes
Hepatic stellate cells	Yes	Yes	Yes
LSECs	Yes	Yes	Yes
Ligands	(T)LCA>DCA>CDCA>CA	CDCA>DCA>LCA>CA	TCA>GCA

LSEC: liver sinusoidal endothelial cells, (T)LCA: (tauro)lithocolic acid, DCA: deoxycholic acid, CDCA: chenodeoxycholic acid, CA: cholic acid, TCA: taurocholic acid, GCA: glycocholic acid.

## Data Availability

Not applicable. Data was not generated as part of this manuscript.

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
