# Peer review of "Bile Acids and Biliary Fibrosis"

_cells, 2023, doi:10.3390/cells12050792_

Round 1

Reviewer 1 Report

Comments to authors:

This is a comprehensive review of the role of bile acids in the development of fibrosis, although much remains to be determined as the authors discuss. The following are a few suggestions:

1.     Line:115 Authors might explain the proposed mechanism of protection from the bicarbonate umbrella, in that alkalizing the bile converts the undercharged protonated bile acids to anions which prevents their diffusing into the bile duct cells where injury could occur.

2.     Line: 156, insert “patho” as it is a pathophysiologic process rather than “physiologic”

3.     Line 225: The quoted studies of ciliated cholangiocytes are from in vitro studies.  It is not clear to this reviewer if this occurs in vitro in rodent or human cholangiocytes. Please clarify and give a reference if there is in vivo data to confirm this observation that not all bile duct cells are ciliated. If so are there regional differences?

4.     Line 272: “The overall effect promotes bile flow in cholestasis”.  What is the evidence for this statement (please reference) or is this speculation as to what might be expected? Please clarify.

5.     Line 365:  “be” should be “by”.

Reviewer 2 Report

The authors present a review of the relationship between bile acids and cholangiocytes with a focus on biliary fibrosis formation. Overall, the review is timely and comprehensive, and the information and insights provided will help the basic scientists and hepatologists in better understanding of the underlying molecular pathogenesis involved in biliary fibrosis development.  The writing is clear, easy to follow and the figures are beautifully created. There is minor focus on bile acid receptor effect on fibrosis in disease. While the authors present a very important topic, there are some concerns that could be addressed prior to recommendation to publication in this journal:

The manuscript could be better written with removal of redundant sentences and detailed article summaries to allow for full comprehension of article content by the readers.

a.      Example: lines 136-137 – could be reworded for clarity

b.      Example: line 269 – the cholangiocytes are part of the liver and are included as external expression of FXR outside of the liver (included with kidney and intestine).

c.      Example: lines 290-293 – the inclusion of OCA is unclear without a transition as to why FXR is a promising target. Additionally, it is unclear what the authors mean by FXR candidates.

d.      Lines 374-375 – more details on this study with RORyT is be necessary. As it stands the sentence is erroneous since the publications mention that RORyT expression is modulated by bile acid receptor activation and not by bile acids themselves. If in vitro or supplemental findings were highlighted in the cited paper (100 and 101) it is recommended that these findings be detailed in this sentence.

e.      Lines 450-455 – The authors have a redundant sentence regarding epigenetic modification/mechanisms and biliary fibrosis/PSC pathogenesis. It is recommended that these be reworded to allow for more concise transition to lnRNA-H19 presentation.

f.       Line 481- is redundant following the former two sentences expressing the same sentiment.

Erroneous statements are presented due to undetailed explanation of mechanisms involved in biliary fibrosis.

g.      Example: line 494- according to studies 100 and 101, RORyT is not described as a bile acid receptor. It is recommended that the authors revisit these studies and clarify findings being summarized in this review.

h.      Example: lines 81-83 – neither citation 8 or 9 describe the cholehepatic shunt. Further, it is recommended the authors find the specific original manuscripts that describe cholehepatic shunting as BA shuttling to hepatocytes through peribiliary capillary plexus and not serum circulation.

i.       Line 66-67: cholangiocytes in PSC undergo hyperplasia and senescence so the statement described in this sentence is unclear. It is recommended that the authors revisit this sentence and add detail to the disease they are describing (I assume PBC).

2.      Detailed review to include references when new topics are being described in lines 247-249 and include relation of Mdr2 to fibrosis in second paragraph of section 3.2.

Minor:

1.      iBAT should be introduced in the first instance of ASBT mention. It is unclear why ileal BA transporter is being introduced when discussing cholangiocytes ASBT. Also, Apical is in the name and describing its location on the apical membrane is an unnecessary detail in this sentence.

2.      Bile acid de novo synthesis is the main end product resulting from cholesterol catabolism not as a function of (lines 73-74)

Reviewer 3 Report

In this review manuscript, the authors focused on bile acid signaling in the pathogenesis of biliary fibrosis. Detailed and up to date summary of link between bile acid/bile acid receptors and biliary fibrosis. This reviewer recommends acceptance of the article upon addressing a few concerns noted below.

1.    It will be better to summarize bile acids composition/bile acid receptor expression (activation) changes in cholangiopathies by table.

2.    Bile acid receptors as drug targets in biliary fibrosis should be expanded.
